# Strategies to Improve Cancer Immune Checkpoint Inhibitors Efficacy, Other Than Abscopal Effect: A Systematic Review

**DOI:** 10.3390/cancers11040539

**Published:** 2019-04-15

**Authors:** Vito Longo, Oronzo Brunetti, Amalia Azzariti, Domenico Galetta, Patrizia Nardulli, Francesco Leonetti, Nicola Silvestris

**Affiliations:** 1Medical Thoracic Oncology Unit, IRCCS Istituto Tumori “Giovanni Paolo II”, Viale Orazio Flacco, 65, 70124 Bari, Italy; vito.longo79@tiscali.it (V.L.); galetta@oncologico.bari.it (D.G.); 2Medical Oncology Unit, Hospital of Barletta, Viale Ippocrate, 15, 70051 Barletta, Italy; dr.oronzo.brunetti@tiscali.it; 3Experimental Pharmacology Laboratory, IRCCS Istituto Tumori “Giovanni Paolo II”, Viale Orazio Flacco, 65, 70124 Bari, Italy; a.azzariti@oncologico.bari.it; 4Pharmacy Unit, IRCCS Istituto Tumori “Giovanni Paolo II”, Viale Orazio Flacco, 65, 70124 Bari, Italy; p.nardulli@oncologico.bari.it; 5Dipartimento di Farmacia-Scienze del Farmaco, University of Bari, Piazza Umberto I, 1, 70121 Bari, Italy; francesco.leonetti@uniba.it; 6Scientific Guidance, IRCCS Istituto Tumori “Giovanni Paolo II”, Viale Orazio Flacco, 65, 70124 Bari, Italy

**Keywords:** immune checkpoint inhibitors, chemotherapy, tyrosine kinase inhibitors, angiogenesis

## Abstract

Despite that the impact of immune checkpoint inhibitors on malignancies treatment is unprecedented, a lack of response to these molecules is observed in several cases. Differently from melanoma and non-small cell lung cancer, where the use of immune checkpoint inhibitors results in a high efficacy, the response rate in other tumors, such as gastrointestinal cancers, breast cancer, sarcomas, and part of genitourinary cancers remains low. The first strategy evaluated to improve the response rate to immune checkpoint inhibitors is the use of predictive factors for the response such as PD-L1 expression, tumor mutational burden, and clinical features. In addition to the identification of the patients with a higher expression of immune checkpoint molecules, another approach currently under intensive investigation is the use of therapeutics in a combinatory manner with immune checkpoint inhibitors in order to obtain an enhancement of efficacy through the modification of the tumor immune microenvironment. In addition to the abscopal effect induced by radiotherapy, a lot of studies are evaluating several drugs able to improve the response rate to immune checkpoint inhibitors, including microbiota modifiers, drugs targeting co-inhibitory receptors, anti-angiogenic therapeutics, small molecules, and oncolytic viruses. In view of the rapid and extensive development of this research field, we conducted a systematic review of the literature identifying which of these drugs are closer to achieving validation in the clinical practice.

## 1. Introduction

Today, immune checkpoint inhibitors (ICIs) represent a gold standard treatment in the first-line setting of several tumors, including non-small cell lung cancer (NSCLC) [1,2,3], BRAF wild-type (WT) melanoma [4] and metastatic renal cell carcinoma (mRCC) [5]. These molecules are antibodies which block checkpoint molecules such as s cytotoxic T-lymphocyte-associated antigen-4 (CTLA-4) and programmed-death1/programmed death-ligand 1 (PD-1/PD-L1). These interferences reduce the immune suppressive mechanisms increasing the immune responses against cancer and can result in tumor regression in many patients. Over time, the CTLA-4 and PD-1 blockade improved overall survival and the survival rate of many tumors, and have been tested in many others with excellent oncological outcomes. Although these molecules appear to be very promising, there are many limitations such as notable side effects (i.e., endocrine failure, gastrointestinal and pulmonary toxicities). Anyway, one of the biggest disadvantages is that ICIs present a lower activity in several cancers, such as those with low mutational burden. Moreover, even for some pathologies where ICIs have a greater activity, there are some patients that do not show any benefit. However, in some cases there is a lack of response to these molecules [6,7,8].

Mainly, two strategies are considered to improve the response rate to ICIs. The first is represented by the selection of patients according to specific predictive factors (i.e., PD-L1 expression, tumor mutational burden (TMB), and clinical features). The second strategy has the aim to enhance the efficacy of ICIs, with the abscopal effect induced by radiotherapy representing the most frequently evaluated approach in both pre-clinical and clinical setting [9,10]. However, in the last few years, several studies were focused on the potential role of molecules as it is able to improve the response rate to ICIs by modifying the immune microenvironment of the tumor, increasing the number of activated T cells exerting effector functions, and decreasing the number of immunosuppressive cells thus transforming a cold tumor into a hot one. These drugs include microbiota modifiers, drugs targeting co-inhibitory receptors, anti-angiogenic therapeutics, small molecules, and oncolytic viruses. A systematic review of the literature was conducted, considering only the drug classes which are under evaluation in the clinical setting and as far as they could be considered in the clinical practice in the near future. The research has been conducted considering papers published on PubMed and data presented to the ASCO and ESMO annual meeting. The aim of this systematic review is to evaluate all the drugs, molecules, and viruses which could improve the activity of ICIs. In particular, we evaluate the immunological mechanisms, which lead to enhance ICIs immune anti-cancer.

## 2. Materials and Methods

### 2.1. Research Strategy

The research strategy was designed to identify published peer-reviewed studies that research the combination of ICIs and other therapies to improve the anti-tumor immune response. The review covered all countries; no time limit has been set to ensure the identification of a wide range of articles. A web-based search of MEDLINE/PubMed library data published from 2010 to December 2018 was performed. Additional research was performed on ClinicalTrials.gov (Figure 1). Search terms were generated to encapsulate the effect of ICIs on cancer and the increase in the antitumor effect (Table 1).

### 2.2. Inclusion/Exclusion Criteria

To be eligible, papers had to be written in English, published in a peer-reviewed journal, be original primary research including experimental, observational, and qualitative studies. 

The relevant outcomes explored were further investigated as there was a demonstrated role for greater efficacy of the ICI anti-cancer effect when these were administrated in combination with other therapies. The authors excluded the use of ICIs in combination with radiotherapies or other local or regional treatments.

### 2.3. Study Selection

All studies identified through the search process were exported to EndNoteversion X7(ClarivateAnalytic 22 Thomson Place, 36T3 Boston, MA, USA). Duplicates were removed. Two authors (O.B. and V.L.) have independently doubled the titles, abstracts and keywords with the eligibility criteria. The results were compared and full-text records of potentially relevant publications were obtained and screened using the inclusion criteria for the final selection of studies for systematic review (Figure 1).

A group of experts provided additional biological and clinical information, greatly helping to clarify some issues in the absence of clear information from the literature.

The final draft was then submitted to expert evaluation and modified according to their suggestions and comments. 

## 3. Microbiota and ICIs

Microbiota plays a crucial role in the development of host immunity [11]. In several pathologies (i.e., inflammatory bowel disease, diabetes, obesity, atherosclerosis, asthma, and dysmetabolic syndromes) gut commensals resulted in being disrupted in comparison with those of unaffected individuals. [12,13]. 

Regarding the relationship between microbiota and ICIs, the administration of a combination of broad-spectrum antibiotics (i.e., ampicillin plus colistin plus streptomycin) as well as imipenem alone compromised antitumor effects of CTLA-4 monoclonal antibody (mAb) as a consequence of microbiota impairment. Moreover, prescription of antibiotics in patients treated with anti–PD-1/PD-L1 mAb between two months before and two months after the start of immunotherapy resulted in a worse prognosis [14], implying a critical role of microbiota in the modulation of response to ICIs.

In 2015 Vetizou et al. [15] found that antitumor effects of anti-CTLA-4 mAb depended on distinct samples of Bacteroides (B). In particular, the authors demonstrated that the specific T cell response for B. thetaiotaomicron and B. fragilis was associated to the efficacy of anti-CTLA-4 administration in mice inoculated with MCA205 sarcomas, Ret melanoma, and MC38 colon cancer cells. Gut bacterial disruption led to a reduction of anticancer response. This deficiency was overcome by administration of B. fragilis, through immunization with B. fragilis polysaccharides, or by adoptive transfer of B. fragilis-specific T cells. Feces transplantation from patients with metastatic melanoma responsive to anti-CTLA-4 in mice inoculated with cancer cells favored the outcome of these mouse tumor models [15]. In fact, the authors re-colonized both germ-free mice and antibiotics treated with bacterial species, finding that B. fragilis, B. thetaiotaomicron, B. cepacia, or the combination of B. fragilis and B. cepacia could restore the anti-CTLA-4 mAb effects. Oral combination of B. fragilis and Burkholderia cepacia could restore the efficacy of CTLA4 blockade in animals treated with antibiotics, without incurring colitis [15,16]. These data were confirmed in a prospective study considering patients with metastatic melanoma treated with ipilimumab. The intestinal microbiome enriched with B. phylum was correlated with a low incidence of checkpoint-block-induced colitis [17].

In another preclinical study, Sivan et al. [18] compared the growth kinetics of B16.SIY melanoma cells subcutaneously inoculated in two genetically similar C57BL/6 mice from Taconic Farms (TAC) and the Jackson Laboratory (JAX) which contained different intestinal bacterial communities [18]. TAC mice generated more aggressive tumors than JAX mice. On the contrary, tumor-infiltrating specific CD8+ T cells were more evident in JAX mice than in TAC mice. The Bifidobacterium genus was identified as a driver of tumor response in JAX mice. When both mice were co-housed, all animals showed a JAX phenotype, suggesting that an enhanced immune response was potentiated by microbes. Moreover, administration of a mixed Bifidobacterium subspecies altered tumor growth in TAC mice [18]. 

Patients with baseline bacterial species with a prevalence of the Faecalibacterium genus and other Firmicutes had a significantly longer progression-free survival (PFS) and overall survival (OS) with a more frequent occurrence of colitis than patients with microbiota characterized by the prevalence of B. Moreover, some of these patients reached an OS longer than 18 months [19].

Another study evaluated the therapeutic efficacy of human intestinal microbiota and its metabolites with different ICIs (i.e., ipilimumab, nivolumab, ipilimumab plus nivolumab, or pembrolizumab). The intestinal microbiota of responder patients was enriched by B. caccae with high levels of anacardic acid. In particular, the bacterial microbiome of patients responsive to the combination of nivolumab plus ipilimumab and pembrolizumab was enriched by Faecalibacteriumprausnitzii, B. thetaiotamicron, Holdemaniafiliformis, and Doreaformicogenerans [20].

In 2018, two parallel studies evaluated the role of the microbiome of melanoma patients treated with anti-PD-1. The first analyzed the oral microbiome of 112 patients without significant differences between responders and non-responders, although the fecal microbiota samples of 30 responders to ICI showed a significant presence of Ruminococcaceae bacteria (*p* < 0.01). In the second study 38 and four patients were treated with anti PD1 and anti CTLA4, respectively. A higher presence of Bifidobacterium longum, Collinsellaaerofaciens, Enterococcus faecium, Bifidobacterium adolescentis, Kleibsiella pneumonia, Veillonellaparvula, Parabacteriodesmerdae, and lactobacillus sp. was observed in the intestines of responders compared to those found in non-responders. Moreover, transplantation of fecal material from responding patients into germ-free mice improved tumor control more effectively than anti-PD-L1 therapy in both studies [21,22].

The microbiome was evaluated in 249 patients with NSCLC, mRCC, and urothelial carcinoma (UC) treated with anti-PD-1 [14]. Genomic analysis of patients’ stool samples revealed a significant correlation between response to treatment and high presence of Akkermansia muciniphila. The antitumor effects of the anti PD-1 blockade were improved when the fecal microbiota from patients with a responding tumor were transplanted into germ-free or antibiotic-treated mice. In contrast, fecal transplantation from patients who did not respond to germ-free mice did not achieve any results. Oral supplementation with Akkermansia muciniphila in these latter mice has restored the efficacy of PD-1. 

Sivan et al. [18] demonstrated greater expression of the major class I and II histocompatibility complex in dendritic cells (DCs) of JAX mice or Bifidobacterium colonized TAC mice. All the above studies showed an influence of the microbiota on DC maturation and activation (Figure 2). An increase in the rate ofCD8+/Treg was observed in mice transplanted with fecal samples from ICI-sensitive patients. The analysis of tumor infiltrates also revealed the increase of innate effector cells and the reduction of myeloid-derived suppressor cells (MDSCs) [21,22]. Moreover, the administration of Akkermansia muciniphila in germ-free mice treated with anti-PD-1was associated with increased frequency of T helper1 Tregs/tumoral helper1 cells. Finally, the oral administration of A. muciniphila and E. hiraestimulates leads DCs to increase the production of IL-12, a cytokine involved in the inhibition of PD-1 under physiological conditions [14]. 

In conclusion, there is evidence supporting the relationship between some bacterial species and the enhanced response to ICIs (i.e., Ruminococaceae family of the Firmicutes phylum as Firmicutesprausnitzii) [19,20,21]. Similarly, other intestine microbiome components (i.e., Bacterioides and Firmicutes phylum) have been associated with a lack of response to immune checkpoint blockade [19,20,21]. Data concerning Firmicutes (Roseburia, Streptococcus) [20,22] and other B. (i.e., Alistipes, Porphyromonaspasteri, and C. aerofaciens) are still not univocal [14,21,22]. It is interesting to note how different bacteria are beneficial in different types of cancer. The differences in the methods used for sample feces and the analysis of the intestinal microbiome, the databases used for analysis, and the populations with both dietary and microbiotic differences are responsible for the ambiguity of these data. These heterogeneous results make it difficult to interpret the reason for such different data. In particular, it is still unclear why different microbiota improve ICI in different types of cancer. Clinical trials are underway to define the possible role of the microbiota in improving the ICI response.

## 4. Chemotherapeutics Sensitizing Tumor to ICIs

Recently, the combination of immunotherapy and chemotherapy has been approved for the treatment of both metastatic and locally advanced NSCLC [1,2,3]. Chemotherapy not only achieves an ulterior efficacy to immunotherapy but it also acts in a synergistic manner in two significant ways: (a) Induction of immunogenic cell death as part of its independent therapeutic effects and(b) disruption strategies used by neoplastic cells to evade immune response. The first process involves the release of tumor antigens and the emission of danger-associated molecular patterns within tumor microenvironment during cell death. At the same time, chemotherapy decreases the number of immunosuppressive cells in the microenvironment including Tregs and MDSCs, increases the number of cytotoxic T lymphocytes (CTLs) and promotes maturation and activation of DCs (Figure 2). In addition, chemotherapeutics modifies the levels of several cytokines, down-regulates immune suppressive cytokines (i.e., transforming growth factor-β (TGF β) and IL10), and up-regulates cytokines promoting tumor immunity (i.e., tumor necrosis factor-α (TNF-α), IL-2, and interferon (IFN)–γ) [23]. 

As far as ICIs are concerned, preclinical models showed that autochthonous tumors that lacked CTLs infiltration were resistant to these agents, while on the contrary the exposure to appropriately selected immunogenic chemotherapeutics induces CTLs tumor infiltration, sensitizing tumor to ICIs. In the mouse model of lung adenocarcinoma, refractory to an anti-PD-1 and anti-CTLA-4 mAb combination therapy, the use of oxaliplatin in combination with a low-dose of cyclophosphamide increased the lung CTLs/Treg cell ratio sensitizing the tumor to ICIs [24]. Similarly, oxaliplatin increased the amounts of CTLs and activated DCs in a murine colorectal cancer, enhancing the efficacy of a PD-L1 trap [25]. A low-dose of cyclophosphamide combined with an anti-PD1 synergistically induced antigen-specific immunity and the infiltration of CD8+ and CD4+FoxP3^−^T cells as well as it induced the suppression of the CD4+ CD25+ FoxP3+ regulatory T cell function, thus resulting in the increase of tumor-free survival in a model of cervical cancer [26,27]. According to these studies, 5-fluoruracil (5-FU) increased tumor immunity in a mouse model by renal cell xenograft through an increase of CTL infiltration mediated by the High Mobility Group Box 1 (HMGB1). Interestingly, a combined 5-FU and anti-PD-L1 treatment significantly improved the relationship between CTL and MDSCs compared to5-FU and anti-PD-L1 single treatments with a longer OS [28]. 

On the other hand, several chemotherapeutics have been shown to induce an up-regulation of PD-L1 expression, as a possible mechanism of chemotherapy immune suppression. However, the increase of PD-L1 expression may support the synergism between chemotherapy and immunotherapy targeting the PD-L1/PD-1 axis. 5-FU demonstrated up-regulation of PD-L1 in two preclinical studies evaluating colorectal cancer patients [25,29]. Similarly, the administration of trabectedin induced the IFN-γ-dependent PD-L1 expression within a tumor in a murine model of ovarian cancer [30]. Others drugs able to up-regulate PD-L1 expression in ovarian cancer models are paclitaxel, carboplatin, cisplatin, gemcitabine, and capecitabine [31].Interestingly, Peng J et al. showed, from a collection of cancer cells from ovarian cancer patients with massive ascites, that the expression of PD-L1 increased 5-fold on day four after combined paclitaxel and carboplatin therapy and decreased to pre-treatment levels on day 11, demonstrating the reversibility of PD-L1 expression induced by chemotherapy [31]. Finally, the evaluation of 150 specimens of patients with ovarian cancer treated with neoadjuvant chemotherapy showed the up-regulation of PD-L1 [32].

Several clinical trials are evaluating the combination of chemotherapy with ICIs, but in the majority of these, chemotherapy and chemotherapeuticsis administered concurrently and at full doses. Only a few trials are focused on the role of chemotherapeutics as sensitizers for immunotherapy, exploring the optimal dose, or the sequence of administration, while preclinical data have shown that these parameters might affect the results.

An open, multi-center, single-arm label, Phase Ib/II, evaluated the daily metronomic dose of 50 mg of cyclophosphamide without interruption of administration, 10 mg/kg of avelumab on day one and every two weeks until progression, and a single fraction of 8Gy radiotherapy in pretreated head and neck cancer patients, showing noun acceptable toxicity [27]. A study concerning metastatic patients with triple negative breast cancer (TNBC) patients investigated induction therapy with various types of chemotherapy [33]. For the induction phase, low doses of chemotherapy were given for two weeks: 50 mg daily cyclophosphamide, twice 40 mg/m^2^ cisplatin or twice 15 mg doxorubicin. Response rates with chemotherapy appear higher in the cohorts where low-dose chemotherapy was used as induction, compared with nivolumab alone. Conversely, an immunotherapy induction phase may also be useful. An induction phase with durvalumab followed by combination therapy of nab-paclitaxel weekly for 12 weeks followed by four cycles of combined therapy with epirubicin and cyclophosphamide was evaluated in patients with TNBC, resulting in a higher pathological CR rate when compared with chemotherapy alone (53.4% versus 44.2%, respectively) [34].

Other trials evaluating the combination of metronomic chemotherapy with ICIs [35] or the impact of chemotherapy on TMB [36] are underway. Future studies should evaluate drugs capable of inducing immune cell death and CTLs tumor microenvironment infiltration, optimizing ICI integration with chemotherapy. 

## 5. ICIs and Antiangiogenic Drugs

The vascular network with its specific components (endothelial cells, pericytes, growth factors, and receptors) plays a key role in the regulation of inflammatory response, wound healing, and immune surveillance. Antigen-primed T cells require a healthy endothelium for the trafficking to tissue districts and the cell-to-cell cross-talk during the priming and effector phase of the immune response. The transit of immune cells in the tumor plays a critical role in the outcome of immunotherapeutic strategies, similarly to classical chemotherapeutic drugs. In particular, a normalized endothelium ensures the correct trafficking of T cells to the tumor bed [37]. In fact, tumor angiogenesis contributes to the escape of the immune tumor through the immunosuppressive activity exerted by VEGF, PGE2, IL-10, and tumor hypoxia. In particular, VEGF acts through both the inhibition of lymphocyte adhesion to activated endothelial cells and the systemic effect on immune-regulatory cell function, including the suppression of DCs maturation, the inhibition of T cell development, and the increase of inhibitory immune cells [38] (Figure 2). Therefore, the possibility of administering ICIs during an anti-angiogenic treatment has been studied in different types of cancers according to the hypothesis that anti-angiogenic drug-induced normalization of the vessels may improve immunotherapeutic strategies. On the other hand, ICI activation of Th1 cells blocked vessel normalization, suggesting the existence of a mutually regulatory circuit [39].

In a phase II study, 46 patients with metastatic melanoma were treated in four dosing cohorts of ipilimumab (3 or 10 mg/kg) with four doses at three-week intervals and then every 12 weeks in combination with bevacizumab (7.5 or 15 mg/kg every three weeks). Eight partial responses and 22 stable diseases were observed, with a disease-control rate of 67.4% and a median OS of 25.1 months [40]. Bevacizumab has been evaluated also in combination with ICI targeting the PD-1/PD-L1 axis in a phase II study considering HER2-negative advanced breast cancer patients. The combination of nivolumab, paclitaxel, and bevacizumab showed an overall response rate ORR of 70% [41]. The same combination demonstrated clinical activity in women with recurrent ovarian cancer, which showed a global confirmed response rate of 21% and a median PFS of 9.4 months [42]. In another study patients with pre-treated NSCLC with platinum-based first-line chemotherapy received nivolumab plus bevacizumab as maintenance therapy with 1-yr OS rate of 75% and a manageable toxicity profile [43]. Recently, the phase III IM power 150 trial showed no new safety signals of the combination of atezolizumab plus bevacizumab, carboplatin, and taxol in first-line non-squamous NSCLC patients with a median OS of 20.5 months [2]. Interestingly, the first randomized phase III trial of a PD-L1/PD-1 pathway inhibitor combined with bevacizumab in first-line mRCC showed longer PFS for atezolizumab plus bevacizumab compared to sunitinb in PD-L1+ patients [44]. The safety and efficacy of a combined treatment of bevacizumab with atezolizumab was assessed in pre-treated patients with metastatic colon rectal cancer (MCRC), or in oxaliplatin-naïve patients in conjunction with FOLFOX (fluorouracil, folinic acid, leucovorin, and oxaliplatin), with an ORR of 44% in the combination group. In a phase 1 a/b study [45] concerning patients with gastric or gastroesophageal junction (G/GEJ), NSCLC, UC, or biliary tract cancer (BTC), the combination of ramucirumab (10 mg/kg) with pembrolizumab (200 mg on the first day of q3w) showed a disease control rate DCR of 85% with no relevant toxicity [46]. Regarding antiangiogenic TKIs, the combination of nivolumab and either pazopanib or sunitinib has been evaluated in mRCC pre-treated with at least one previous systemic therapy. An 45% ORR was demonstrated in the nivolumab plus pazopanib arm, compared to 52% in the nivolumab plus sunitinib arm, with a manageable safety profile. These combination approaches might benefit patients with poor prognosis, such as those with a low probability to respond to ICI monotherapy (i.e., refractory to patients on first-line therapy or showing PDL1–negative tumors) [47]. Considering the potential role of antiangiogenenic therapies of changing a cold tumor into a hot one, several trials are currently underway investigating other combinations of antiangiogenic agents and ICIs.

## 6. Strategies Involving Other Co-Inhibitor Receptors

The encouraging outcome obtained by the co-inhibitory receptors CTLA-4 and PD-1 prompted the research of additional co-inhibitory molecules. T cell immunoglobulin and immune-receptor tyrosine-based inhibitory motif domain (TIGIT) is a newly identified co-inhibitory receptor expressed by Tregs, activated T cells, and natural killer (NK) cells [48]. TIGIT expression is elevated on CD8^+^TILs and Tregs in a variety of tumors, as well as the expression of its three ligands, namely CD155, CD112, and CD11 3 [49]. Moreover, TIGT and PD-1 are co-expressed and up-regulated on TILs. Dual blockade of two immune checkpoints enhances function of TILs resulting in a significant tumor rejection, as demonstrated by the combination of anti-CTLA-4 with anti-PD-1/PD-L1.Anti-PD-1 and anti-TIGT dual therapy significantly improved survival compared to control and monotherapy in a murine glioblastoma (GBM) model. Clinically, TIGIT expression on tumor-infiltrating lymphocytes was shown to be elevated in GBM samples, suggesting that the TIGIT pathway may be a valuable therapeutic target [50]. A phase II, randomized, blinded, placebo-controlled trial is currently underway that considersMTIG7192A, an anti-TIGIT antibody, in combination with atezolizumab patients with chemotherapy-naïve NSCLC [51]. 

Lymphocyte activation gene-3 (LAG3), an immune checkpoint up-regulated on activated T cells, Treg, and NK cells in different types of cancer, is required for the maintenance of Treg suppressive function. LAG3 blocks either by soluble LAG3 immunoglobulin or antibodies have shown efficacy in the antitumor response. Similarly, to TIGT, LAG3 coexisted and upregulated with PD-1 on TILs [52]. According to preclinical data showing a significant increase in the activity of dual blockade of LAG3 and PD-1 [53], numerous clinical trials are underway with the aim of translating this combination modality into clinical practice. 

An open-label phase 1/2a trial evaluating BMS-986016, an experimentalanti-LAG-3, in combination with nivolumab in patients with advanced melanoma previously treated with anti-PD-1/PD-L1 therapy (*n* = 55). ORR was 12.5% in evaluable patients (*n* = 48). The expression of LAG-3in at least 1% (*n* = 25) of tumor-associated immune cells within the tumor margin was associated with an almost triple improvement in ORRs compared to patients without LAG-3 expression (*n* = 14) (20% and 7.1%, respectively) [54]. LAG525, a humanized IgG4 mAbs capable of blocking the binding of LAG-3 to class IIMHC, is being studied in a phase I/II study in combination with an anti-PD1 treatment. Common adverse events (≥10%) were fatigue (10%) for LAG525 alone and fatigue (18%), diarrhea (15%), and nausea (12%) in the combination group. LAG525 plus the anti-PD1 spartalizumab drug led to durable RECIST responses (11 PR, 1 CR) in a variety of solid tumors, including mesothelioma (2/8 pts) and triple-negative breast cancer (TNBC) (2/5 pts). In TNBC tumor biopsies, a tendency to convert immuno-cooled biomarker profiles to immune-activated has been reported [55]. T cell immunoglobulin containing the mucin domain 3 (TIM-3) is widely expressed on helper 1 T cells, CD8+ lymphocytes, Treg, DCs, NK cells, and monocytes. Similarly, to TIGIT and LAG3, the high expression of TIM-3 and PD-1 is observed in the tumor microenvironment, in particular on TIL and Treg, suggesting the possible re-establishing of T cell function through the targeting of TIM-3 and PD-1 [56]. A phase 1 study is evaluating the anti-TIM-3 antibody (T cell immunoglobulin and protein-containing mucin 3) TSR-022 as monotherapy and in combination with an anti-PD-1 antibody, in pre-treated patients with advanced solid tumors [57].

## 7. Oncolytic Virus and ICIs

The oncolytic virus vectors are designed to have a high tumor tropism, maximize cancer killing effects and minimized a mage to surrounding normal tissue. It is interesting to note that these viruses not only facilitate the lysis of tumor cells, but also cause a strong change in the tumor immune microenvironment. In particular, oncolytic viruses transfer the genes encoding IFN-α, GM-CSF, and others cytokines that induce tumor-specific immunity by promoting DC maturation and function. On the other hand, tumor cell lysis induced by oncolytic viruses determines the release of damage-associated molecular patterns (DAMPS) that include cell surface proteins, membrane proteins, and nucleic acids (Figure 3) [58,59].

Talimogene Laherparepvec (T-VEC) replicates within tumors and produces GM-CSF, resulting in a first-rate FDA-approved intralesional oncolytic immune therapy for stage IIIb and IV melanoma. A recent phase II trial comparing ipilimumab plus T-VEC with ipilimumab alone showed that the ORR in the combination arm was significantly higher than in the monotherapy arm (39% vs. 18%; *p* < 0.02) [60]. Distant non-injection sites demonstrated an adjuvant effect with a reduction in visceral lesions size in 52% of patients in the combination arm versus only 2% of the patients in the ipilimumab arm. T-Vec has been also tested in patients with melanoma in combination with pembrolizumab in a phase Ib study. No dose-limiting toxicity was observed with an ORR of62% and a CRR of 33% [58]. A phase III trial is underway [61]. It is interesting to note that an analysis performed prior to the administration of anti-PD1 antibodies showed that T-VEC increased the PD-L1 expression and inflammation distant from the injection sites. HF 10, another virus included in the HSV family, in combination with ipilimumab showed in a phase II clinical trial regarding stage IIIB/IIIC or IV unresectable melanoma a DCR of 68% without disease limiting toxicity [62]. An oncolytic adenovirus competent for replication with tumor selectivity, namely Tasadenoturev (DNX-2401), was able to overcome the exhaustion of T cells demonstrating a reduction in tumor size in a phase I study for patients with recurrent GBM. A phase II study employing DNX-2401 and pembrolizumab in GBM progressed after initial therapy is currently underway [63]. Another group of oncolytic viruses is represented by vaccinia viruses, members of the Poxviridae family, which are suitable for transgene insertion. Pexa-Vec targeted tumor-associated endothelial cells resulting in vascular disruption and oncolysis [64]. A single dose of Pexa-Vec intravenously demonstrated activation of NK, CD4/CD8 T cells, and antigen presenting cells in surgically treated liver metastases. The combination of Pexa-Vec and nivolumab is under investigation for the treatment of liver tumors [65]. Furthermore, the combination of Pexa-Vec with other ICIs is being evaluated in colorectal cancer and other advanced tumors, respectively [66,67]. When speaking about Coxackieviruses, CVA 21 is able to increase infiltration of immune cells and checkpoint molecules, several clinical trials concerning the combination of CVA 21 with ICIs are ongoing [68]. In particular, in the CAPRA clinical trial, patients who receive multiple intratumoral injections of CVA 21 and pembrolizumab showed an ORR of 73% [69]. Finally, the reoviruses, characterized by icosahedral capsid and double-stranded RNA genomes, have been shown to increase cytotoxic T cells infiltrating the CD8+tumor in an Ib phase concerning GBM patients undergoing debulking neurosurgery [70]. A clinical trial [71] is underway evaluating the use of a reovirus, namely pelareorep in combination with pembrolizumab and chemotherapy in patients with recurrent metastatic pancreatic cancers.

## 8. Small Molecule Inhibitors and ICIs

Various evidence suggests that small molecule inhibitors could improve host-tumor interactions, improving antigen expression and the immune response against tumor cells [72]. Several small molecules in combination with ICIs have been studied for the treatment of different types of tumor histotypes (Table 2).

The first combination of small molecule inhibitors and ICIs have been evaluated in melanoma. In particular, since the administration of BRAFi/MEKi represents a standard treatment of metastatic BRAF^V600E^ melanoma, the possibility that this association would be improved by ICIs has been evaluated. It has been demonstrated that BRAF inhibition is associated with enhanced melanoma antigen expression [73,74,75]. Moreover, selective BRAF inhibitors induce marked T cell infiltration in human metastatic melanoma [74], with an up-regulation of PD-L1 in the tumor microenvironment [72,74]. Nevertheless, the benefit of this combination in preclinical models has been modest [76,77,78,79]. In particular, in a mouse model of syngeneic BRAF^V600E^ driven melanoma, the combination of dabrafenib and trametinib with pmel-1 adoptive cell transfer showed a complete tumor regression with increased T cell infiltration in tumors and improved in vivo cytotoxicity. Single agent dabrafenib increased the number of tumor-associated macrophages and Tregs in tumors that conversely decreased with the addition of trametinib. The combination of BRAFi/MEKi and ICI induced either an increased expression of the melanosomal antigens and MHC or the global immune-related gene up-regulation. Moreover, a combination of dabrafenib and trametinib with anti-PD1 therapy in SM1 tumors led to a greater anti-tumor effect compared to the results obtained with the only small molecules combination [80].

The first phase 1 trial evaluating the role of ipilimumab in combination with vemurafenib was stopped after one month due to liver toxicity [81]. Another phase 1 study evaluated the safety of the combination of dabrafenib, trametinib, and ipilimumab. This study was also stopped due to excessive colon toxicity [82]. A combination of dabrafenib and ipilimumab demonstrated an ORR of 69% in the 26 BRAF-mutated patients with a good safety profile [83]. The KEYNOTE-022, an ongoing phase I/II trial [98], is evaluating the combination of pembrolizumab with dabrafenib and trametinib. Preliminary data on 15 patients enrolled across dose determination and dose confirmation arms showed a safety profile and an ORR of 60% (*n* = 9 PR, *n* = 2 SD, *n* = 3 PD) [85]. A phase Ib study is investigating vemurafenib and atezolizumab combination and comparing this combination concurrently or after a run-in period with vemurafenib alone [99]. It was demonstrated that the vemurafenib run-in showed a higher ORR than concomitant atezolizumab plus the onset of vemurafenib. The combination of atezolizumab, vemurafenib, and cobimetinib in this subgroup of patients is being investigated [84]. Preliminary results confirmed that this combination has a manageable safety profile with a promising antitumor activity in patients with BRAF^V600^-mutant metastatic melanoma [86].

Also, in gastrointestinal stromal tumors (GIST), preclinical studies demonstrated that imatinib in combination with ICIs should improve the immune response. It is well known that this drug induces NK cells activity through DCs in several cancers [100,101]. Furthermore, in an in vitro study, imatinib reduced the Treg immunosuppressive function and the FoxP3 expression with the inhibition of phosphorylation of both ZAP70 and LAT, impairing their immunosuppressive function [87]. Moreover, PFS correlated with IFN-γ secretion by NK cells in patients affected by GIST treated with imatinib [88]. In a mouse model of spontaneous GIST, Balachandran et al. demonstrated that the immune system substantially contributed to the anti-tumor effects of imatinib. In fact, it activated CD8+ T cells and induced Treg apoptosis in the tumor sample by reducing immunosuppressive enzyme indoleamine 2,3-dioxygenase (IDO) [89]. In a more recent study, PD-1 was expressed more on T cells in imatinib-treated human GISTs as compared to untreated patients. Imatinib inhibited the upregulation of PD-L1 through IFNγ in human GIST cell lines. In a GIST mouse model, imatinib down-regulated IFNγ related genes and reduced the PD-L1 expression on tumor cells. Moreover, PD-1 or PD-L1 blockade without imatinib achieved no response in GIST mouse model. On the contrary, association of ICIs and imatinib increased antitumor effects by enhancing cytotoxic T cell effector function [90].

A current phase I study is evaluating the effect of a combination of ipilimumab and imatinib GIST positive solid cancers and other-Kit [91]. Preliminary results have shown that this combination is safe on most types of tumors. Nevertheless, low activity without a clear synergy signal is observed in GIST expansion or escalation cohorts [92].

It is interesting to note that a combination of small molecules and ICIs have been evaluated in a mouse model of oral cancer. In this neoplasia, both activation of PI3K/mTOR and MEK/ERK pathways promoted the immunosuppressive tumor microenvironment [102]. In an immunogenic model of cancer of the oral cavity, rapamycin reduced tumor growth in a CD8-dependent manner [93]. More recently, Moore et al. [94] demonstrated that rapamycin improved IFNγ production by peripheral and tumor-infiltrating CD8 T cells in a mouse model of oral cancer. Furthermore, antitumor efficacy was enhanced by the CD8 T cell but not by NK cell. Non-inflamed tumor models, which represent the low level of response to immune therapies, did not induce T cell or NK CD8 cell–mediated antitumor immunity when treated with combinations of targeted and ICIs. In other models, antitumor immune responses to PD-L1 mAb treatment were enhanced when treated with mTOR inhibitors. These data suggested that a combination of mTOR and ICIs inhibitors should be evaluated in clinical trials setting.

There are few preclinical studies considering small molecules inhibitors and ICIs combinations in breast cancer patients. In both murine models and breast cancer patients, CDK4/6 inhibition induced anti-tumor immunity through suppression of Tregs and contributing to anticancer effects [95]. Since cyclin D-CDK4 regulated PD-L1 protein expression, inhibition of CDK4/6 in vivo increases PD-L1 protein levels through inhibition of cyclin D-CDK4. Combination of CDK4/6 inhibitor and anti-PD-1 immunotherapy enhanced tumor regression and dramatically improved OS rates in mouse breast cancer models [96]. Teo et al. demonstrated that PI3K antagonist and CDK4/6 inhibition significantly increased tumor immunogenicity through generating immunogenic cell death in triple negative breast cancer model. Moreover, this combination significantly increased tumor-infiltrating T cell activation and cytotoxicity with reduction of immune-suppressive myeloid-derived suppressor cells. Association of immune checkpoints PD-1, CTLA-4 to PI3K antagonist and CDK4/6 inhibition induced complete and durable regressions (> one year) of breast tumors in in vivo models [97].

In the era of precision medicine, several small molecules have been demonstrated to be active in targeting specific pathways leading to apoptosis of cancer cells with impressive results in anti-cancer treatment. In addition, these molecules appear capable of increasing tumor immunogenicity through the increase of cancer antigens and the activation of cytotoxic activity of CD8 cells leading to an increased putative activity of ICIs when associated in concomitant or sequential therapeutic schedules.

## 9. Conclusions

This systematic review has summarized the current study of the main classes of drugs which improve the activity of the ICIs. The assessment of drugs able to modify the tumor immune microenvironment in addition to ICIs is a field of research, which is currently undergoing a significant escalation. Despite the encouraging results, only chemotherapy has currently adhered to clinical practice for this specific use. Curiously, most of these molecules are characterized by a high level of safety and already consolidated clinical use for indications other than those considered in this study. These features should allow for the possibility of undertaking more extensive and well-designed studies. At the same time, the possibility of new side effects due to the combinatorial strategies or the potential amplification of the well-known ICIs side effects [103,104] should be carefully monitored.

## Figures and Tables

**Figure 1 cancers-11-00539-f001:**
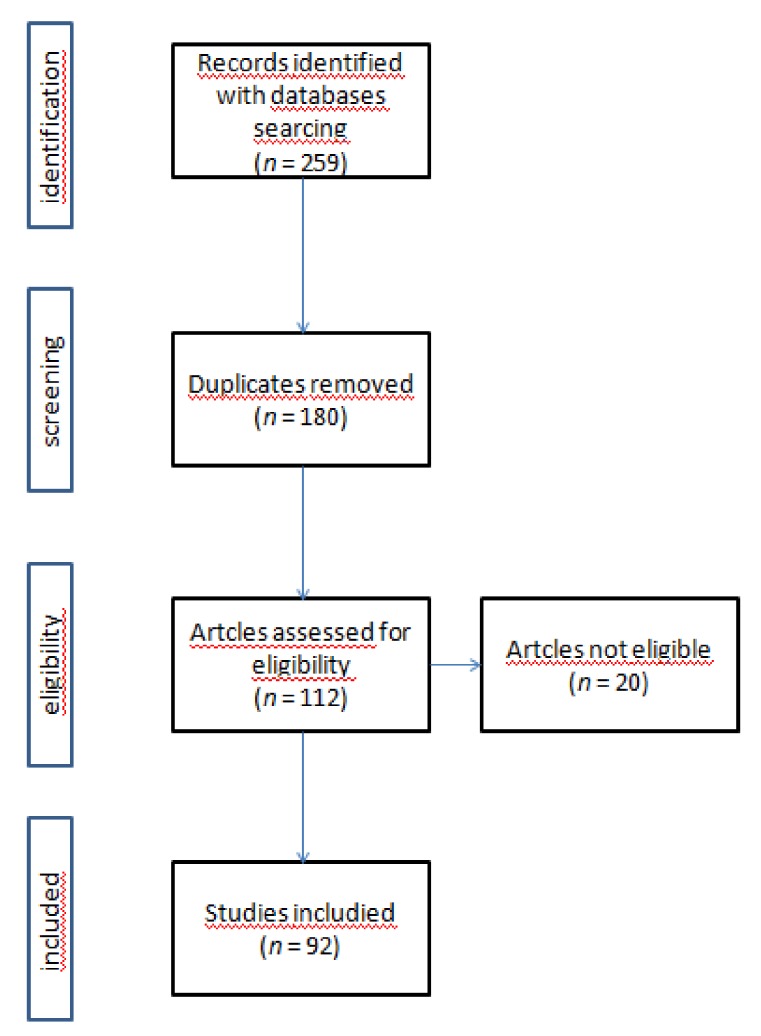
Research strategy with PRISMA flow diagram.

**Figure 2 cancers-11-00539-f002:**
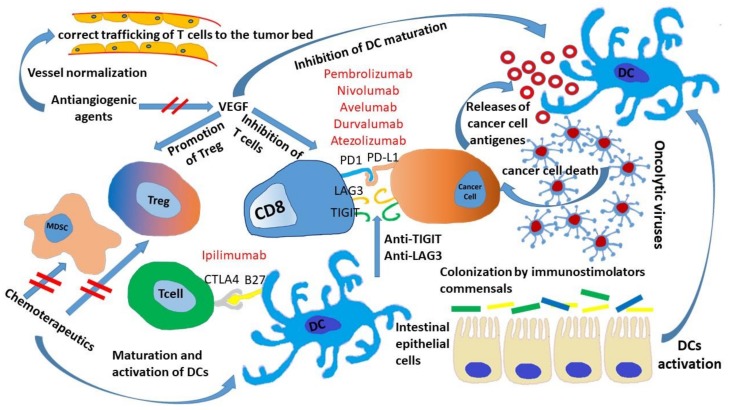
Summarizes the mechanisms implicated in improving the efficacy of immune checkpoint inhibitors (ICIs): The influence of microbiota on dendritic cell (DC) maturation and activation; the correct trafficking of T cells to the tumor bed due to the normalization of endothelium by anti-angiogenic drugs and the VEGF immunosuppressive activity; the impact of chemotherapy on immunosuppressive cells and on DC maturation; release of damaged molecular patterns after oncolytic viruses induce tumor cell lysis.

**Figure 3 cancers-11-00539-f003:**
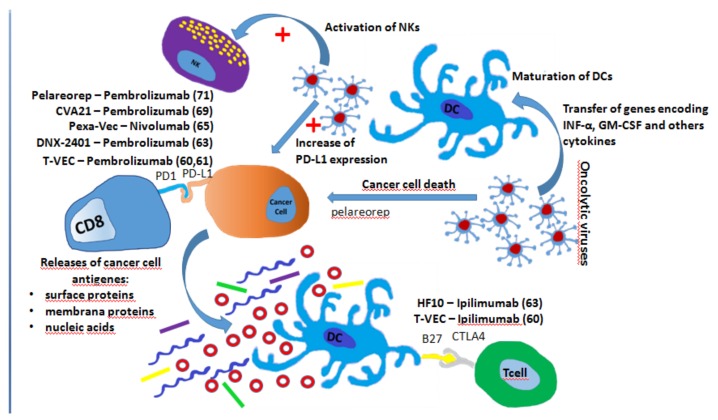
Summary of the mechanisms involved in improving of ICI efficacy by oncolytic viruses: Release of damage-associated molecular patterns after tumor cell lysis, transfer of genes encoding INF-α, GM-CSF and others cytokines, DC maturation and activation, natural killer (NK) cell activation, and increase in PD-L1 expression. The main studies evaluating the combination of oncolytic viruses are also reported.

**Table 1 cancers-11-00539-t001:** Search terms.

Immune Therapy	Enhancer
‘immune checkpoint inhibitors’, ‘anti-PD-(L)1’, ‘anti-CTLA-4’.	“microbiote” OR “microbiota” OR “gut microbe” OR “bacteria”
‘immune checkpoint inhibitors’, ‘anti-PD-(L)1’, ‘anti-CTLA-4’.	“chemotherapy” OR“ chemotherapeutics” OR “metronomic chemotherapy”
‘immune checkpoint inhibitors’, ‘anti-PD-(L)1’, ‘anti-CTLA-4’.	“anti-angiogenetic therapies” OR “bevacizumab” OR “nintedanib” OR “Aflibercept” OR “pazopanib” OR “sunitinib”
‘immune checkpoint inhibitors’, ‘anti-PD-(L)1’, ‘anti-CTLA-4’.	“co-inhibitor receptors” OR “TIGIT” OR “LAG3” OR “TIM-3”
‘immune checkpoint inhibitors’, ‘anti-PD-(L)1’, ‘anti-CTLA-4’.	“Oncolytic virus” OR “adenovirus” OR ”vaccinia viruses” OR ”Coxsackieviruses” OR ”Reoviruses”
‘immune checkpoint inhibitors’, ‘anti-PD-(L)1’, ‘anti-CTLA-4’.	“small molecules” OR “tyrosine kinase inhibitor” OR “mTOR inhibitor” OR “cyclin inhibitor”

**Table 2 cancers-11-00539-t002:** Small Molecule Inhibitors and ICIs.

Small Molecule Enhancer	ICI	Cancer	Study Design	Results/Enhancing	Reference
BRAFi	Not associated	Melanoma	In vitro	BRAF inhibition enhance melanoma antigen expression	Wilmott, 2013. [72]
Selective BRAF inhibitors	Not associated	Melanoma	In vitro	Induction of Tcell infiltration into human metastatic melanoma Up-regulation of PD-L1 in tumor microenvironment	Wilmott, 2012. [74]
Dabrafenib and trametinib	(pmel-1 adoptive cell transfer)	BRAFV600E driven melanoma	In vivo—mouse model	Complete tumor regression with increased T cell infiltration into tumors and improved in vivo cytotoxicity	Cooper, 2014. [79]
Dabrafenib and trametinib	anti-PD1	SM1 tumors (melanoma)	In vivo—mouse model	Superior anti-tumor effect compared to the results obtained with the only small molecules combination	Hu-Lieskovan, 2015. [80]
Vemurafenib	Ipilimumab	Melanoma	Phase 1 trial	Stopped after one month due to liver toxicity	Ribas, 2013. [81]
Dabrafenib, trametinib	Ipilimumab	Melanoma	Phase 1 trial	Stopped due to excessive colon toxicity	Minor, 2015. [82]
Dabrafenib	Ipilimumab	BRAF-mutated melanoma	Phase 1 trial	ORR of 69% Good safety profile	Puzanov, 2014. [83]
Dabrafenib and trametinib	pembrolizumab	BRAF-mutated melanoma	KEYNOTE-022, an ongoing phase I/II trial	ORR of 60% (*n* = 9 PR, *n* = 2 SD, *n* = 3 PD)	NCT02130466, [84] Ribas, 2016. [85]
Vemurafenib (V)	Atezolizumab (A)	Melanoma	Phase Ib trial (V-run in vs. concurrent V-A)	Higher ORR was seen with V run-in than with concurrent A + V start	Sullivan, 2016. [86]
Vemurafenib, and cobimetinib	Atezolizumab	BRAF^V600^-mutant melanoma	Phase I/II trial	Manageable safety profile and promising antitumor activity	NCT01656642. [84]
imatinib	Not associated	GIST	In vitro study	Reduction of Treg immunosuppressive function	Larmonier, 2008. [87]
Imatinib	Not associated	GIST	In vivo study	PFS correlated with IFN-γ secretion by NK cells	Ménard, 2009. [88]
Imatinib	Not associated	GIST	In vivo—mouse model	Activated CD8+ T cells and induced Treg apoptosis in tumor sample	Balachandran, 2011. [89]
Imatinib	Anti-PD-1 (RMP1-14) or anti-PD-L1 (10F.9G2)	GIST	In vivo—mouse model	Increased antitumor effects by enhancing cytotoxic T cell effector function	Seifert, 2017. [90]
Imatinib	Ipilimumab	GIST and other c-Kit positive solid cancers	Phase 1 trial	Manageable safety profile in multiple tumor types. Low activity with no clear signal for synergy in escalation or GIST expansion cohorts	NCT01738139, [91] Reilley, 2017. [92]
Rapamycin	Not associated	Oral cancer	In vivo—mouse model	Reduction of tumor growth through CD8-activity	Cash, 2015. [93]
Rapamycin	Not associated	Oral cancer	In vivo—mouse model	Enhancing of IFNγ production by peripheral and tumor-infiltrating CD8 T cells	Moore, 2016. [94]
Rapamycin	PD-L1 mAb	Oral cancer	In vivo—mouse model	Activation of CD8 T cells in tumor infiltration increased by the addition of rapamycin	Moore, 2016. [94]
CDK4/6 inhibitor	Not associated	Breast cancer	In vivo—mouse model/blood sample patients	Anti-tumor immunity through proliferation of Tregs	Goel, 2017. [95]
CDK4/6 inhibitor	anti-PD-1	Breast cancer	In vivo—mouse model	Enhancing of tumor regression and dramatically improving of OS	Zhang, 2018. [96]
CDK4/6 inhibitor and PI3K antagonist	Anti PD-1 and anti CTLA-4	Triple negative breast cancer	In vivo—mouse model	Inhibition induced complete and durable regressions (> one year) of breast tumors in in vivo models.	Teo, 2017. [97]

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
