# Peer review of "Strategies to Improve Cancer Immune Checkpoint Inhibitors Efficacy, Other Than Abscopal Effect: A Systematic Review"

_cancers, 2019, doi:10.3390/cancers11040539_

Reviewer 1 Report

This systematic review has recapitulated the reports of drugs that improve response and efficacy of ICIs. The review is well written and has raised some key points such as the effect of microbiata, angiogenic factors, immune-cell-death induced by chemotherapeutics and oncolytic virus, and tumor immune microenvironment in regulating the activity of ICIs. This is a timely topic and this review provides a comprehensive overview of the most important publications of researches aiming at enhancing the efficacy of ICIs.

For improvement:

-       Line 301, correct TIGT to TIGTIT

-       438: the statement regarding one of the mechanisms of CDK4/6I on tumor inhibition in breast cancer models  (ref 99) is not on enhancing T reg cells proliferation but is the suppression of Tregs. Please correct

-        Line 458: “Despite the encouraging results, none of these molecules has currently entered clinical practice for this specific use. Intriguingly, most of these molecules are characterized by a high level of safety and an already consolidated clinical use for indications other than those considered in this study” Please check this statement that is in discrepancy with studies and clinical trials showing increased efficacy of ICIs with chemotherapeutics, as described in section 4.

 Author Response

Thank you very much for the congratulations and for required revisions, useful to improve the manuscript. 

Responses to rev. 1

-       Line 301, correct TIGT to TIGTIT

The Authors corrected this and other misspellings

-       438: the statement regarding one of the mechanisms of CDK4/6I on tumor inhibition in breast cancer models  (ref 99) is not on enhancing T reg cells proliferation but is the suppression of Tregs. Please correct

The Authorsmodified this mistake in the text

-        Line 458: “Despite the encouraging results, none of these molecules has currently entered clinical practice for this specific use. Intriguingly, most of these molecules are characterized by a high level of safety and an already consolidated clinical use for indications other than those considered in this study” Please check this statement that is in discrepancy with studies and clinical trials showing increased efficacy of ICIs with chemotherapeutics, as described in section 4.

The Authors corrected the sentence in the text. In particular, they wrote “The assessment of drugs able to modify tumor immune microenvironment in addition to ICIs is a field of research which is currently undergoing a significant escalation. Despite the encouraging results, only chemotherapeutics have currently entered clinical practice for this specific use.

Reviewer 2 Report

The manuscript by Longo et al., is a comprehensive review of the literature on the combination of immune check point inhibitors (ICI) with other agents such as microbiota, small molecules, oncolytic viruses etc. I have a few minor comments:

1) While the title has the word ‘abscopal effect’ in the context of ICI, no references were provided to point towards literature evidence of abscopal effect of ICI when combined with radiotherapy. The authors should consider including the reference- Chuong et al., J Clin Oncol Res 2017, 5(2):1058, in introduction section.

2) There is emerging evidence that ICI causes acute exacerbation of inflammatory responses leading to conditions such as third-degree heart block and cardotoxicity. The authors should include a paragraph on this important side effect and elude the potential synergistic toxicity with other combination therapies mentioned by the authors (possibly after section 8). They could refer to the following reference: Lancet (2018) 391:933; Tajiri and Ieda Front. Cardiovasc. Med. 2019, 6:3.

3) There are several spelling mistakes across the article, and I think the authors should get this proof-read by English expert. Just to point a few:

(i) line 27 : ‘high sensibility’ should re-written I think the authors meant to  say ‘…higher expression of immune inhibitory molecules…’

(ii) line 36; ‘Tirosin’ should be ‘Tyrosine’

(iii) Table 1 last line ‘cycline’ should be ‘cyclin’

(iv) line 185: ‘INF” should be ‘IFN’… as this is the acronym used by them in line 428.

(v) line 186 ‘autoctonous’ should be ‘autochthonous’

Author Response

Thank you very much for the congratulations and for required revisions, useful to improve the manuscript. 

Responses to rev. 2

1) While the title has the word ‘abscopal effect’ in the context of ICI, no references were provided to point towards literature evidence of abscopal effect of ICI when combined with radiotherapy. The authors should consider including the reference- Chuong et al., J ClinOncol Res 2017, 5(2):1058, in introduction section.

The Authors added the reference to the text

2) There is emerging evidence that ICI causes acute exacerbation of inflammatory responses leading to conditions such as third-degree heart block and cardiotoxicity. The authors should include a paragraph on this important side effect and elude the potential synergistic toxicity with other combination therapies mentioned by the authors (possibly after section 8). They could refer to the following reference: Lancet (2018) 391:933; Tajiri and Ieda Front. Cardiovasc. Med. 2019, 6:3.

Thank you very much for the important observations regarding the new potential safety issues by combination therapy with ICIs. Unfortunately, unlike the abscopaleffect in which new safety issues are already reported in the literature, data on possible side effects resulting from these new combinatorial strategies are still under investigation. In any case, we have included your pertinent reflection and the suggested literature references in the conclusion of the manuscript.

3) There are several spelling mistakes across the article, and I think the authors should get this proof-read by English expert. Just to point a few:

(i) line 27 : ‘high sensibility’ should re-written I think the authors meant to  say ‘…higher expression of immune inhibitory molecules…’

(ii) line 36; ‘Tirosin’ should be ‘Tyrosine’

(iii) Table 1 last line ‘cycline’ should be ‘cyclin’

(iv) line 185: ‘INF” should be ‘IFN’… as this is the acronym used by them in line 428.

(v) line 186 ‘autoctonous’ should be ‘autochthonous’

The Authors corrected the all the misspellings. The manuscript has been revised to a native English speaker.

Reviewer 3 Report

The authors have done a comprehensive review of drugs used in combination with immune checkpoint inhibitors to treat cancers. The review talks about a lot of studies and is very detailed. 

The introduction needs to be expanded more to include some previous knowledge about checkpoint inhibitors. The authors can talk about history of checkpoint inhibitors, effects seen with using just the inhibitors without combining them, advantages and disadvantages of using ICIs and why there is a need to use ICI in combination with other drugs. It is important to tell the readers why you are writing this review and the authors should do that here. 

The end of the introduction should include a few lines on what readers can expect to read in the review. 

The authors list a lot of interesting studies about microbiota and ICI. Is there a reason why different bacteria are advantageous in different cancers? The authors should interpret the results observed in different studies.

The section about oncolytic virus is very interesting. It will be useful to readers to have a cartoon/chart about the different oncolytic viruses used and the drugs used in combination with each virus.

Author Response

Thank you very much for the congratulations and for required revisions, useful to improve the manuscript. 

Responses to rev.3

The introduction needs to be expanded more to include some previous knowledge about checkpoint inhibitors. The authors can talk about history of checkpoint inhibitors, effects seen with using just the inhibitors without combining them, advantages and disadvantages of using ICIs and why there is a need to use ICI in combination with other drugs. It is important to tell the readers why you are writing this review and the authors should do that here.

The end of the introduction should include a few lines on what readers can expect to read in the review.

The Authors revised and rewrote several parts of introduction

The authors list a lot of interesting studies about microbiota and ICI. Is there a reason why different bacteria are advantageous in different cancers? The authors should interpret the results observed in different studies.

There are few studies on microbiota and ICIs. Moreover these studies need to be better improved in more large cohorts with a standardization of methods, which result heterogeneous, just  as the Authors described. With these premises, it is very difficult to interpret these results, so the Authors inserted a sentence in the text explaining this difficulty. 

The section about oncolytic virus is very interesting. It will be useful to readers to have a cartoon/chart about the different oncolytic viruses used and the drugs used in combination with each virus.

The Authors added a cartoon for oncolytic viruses section.